# Manipulation of Alcohol and Short-Chain Fatty Acids in the Metabolome of Commensal and Virulent *Klebsiella pneumoniae* by Linolenic Acid

**DOI:** 10.3390/microorganisms8050773

**Published:** 2020-05-21

**Authors:** Ryan Yuki Huang, Deron Raymond Herr, Shabbir Moochhala

**Affiliations:** 1Canyon Crest Academy, San Diego, CA 92130, USA; rhuang53@gmail.com; 2Department of Mechanical and Aerospace Engineering, University of California, San Diego, CA 92093, USA; 3Department of Pharmacology, National University of Singapore, Singapore 117600, Singapore; phcdrh@nus.edu.sg

**Keywords:** *Klebsiella pneumoniae*, linolenic acid, metabolome, microbiome, prebiotics

## Abstract

Endogenous alcohol produced by the gut microbiome is transported via the bloodstream to the liver for detoxification. Gut dysbiosis can result in chronic excess alcohol production that contributes to the development of hepatic steatosis. The aim of this study was to examine whether linolenic acid can manipulate the production of harmful alcohol and beneficial short-chain fatty acids (SCFAs) in the metabolome of commensal *Klebsiella pneumoniae* (*K. pneumoniae*) and the virulent *K. pneumoniae* K1 serotype. Glucose fermentation by the *K. pneumoniae* K1 serotype yielded increased production of alcohol and decreased SCFAs (especially acetate and propionate) compared to those of commensal *K. pneumoniae*. However, the use of linolenic acid instead of glucose significantly reduced alcohol and increased SCFAs in the fermentation media of the *K. pneumoniae* K1 serotype. The work highlights the value of shaping the microbial metabolome using linolenic acid, which can potentially regulate the gut–liver axis for the prevention and treatment of alcohol-induced liver diseases.

## 1. Introduction

The human gut microbiome consists of the entirety of the microbes that reside in the human gut [1]. These microbes aid in human innate immunity, metabolic functions, and overall wellbeing. However, dysbiosis in the gut microbiome can alter host–microbe interactions and yield detrimental effects for the host. The gut microbiome links and modulates the activity of other human organs through its axes, such as the gut–brain axis, the gut–lung axis, and the gut–liver axis. Thus, manipulation of the biological functions of microbes to alter the production of microbial metabolites in the gut can extend beyond local effects on the gut and may influence distant organ systems.

Certain bacteria residing in the human gut are associated with liver damage via the gut–liver axis. For instance, the high alcohol-producing W14 and TH1 strains of Gram-negative *Klebsiella pneumoniae* (*K. pneumoniae*) are proven to be major factors in contributing to non-alcoholic fatty liver diseases (NAFLD) [2]. Cytolysin-producing *Enterococcus faecalis* (*E. faecalis*), a Gram-positive bacterium, has been found in larger quantities in alcoholic hepatitis patients than their healthy counterparts [3]. Most notably, hypervirulent strains of the capsular serotypes K1 or K2 of *K. pneumoniae* are known to induce liver abscesses or bacteremia in patients. The literature has demonstrated that the *K. pneumoniae* K1 serotype in particular can induce primary liver abscesses and is a risk factor for metastasis [4]. The serotype has been shown to exhibit large amounts of a hypermucoviscous phenotype that enhances its ability to cause severe infections. There are two genes that are associated with its virulence. The first is the mucoviscosity-associated gene A (*magA*). The *magA* gene enables the ability to form a mucoviscous string and to obtain higher resistance to phagocytosis, which has been widely reported in Asia. The second virulence gene is the regulator of the mucoid phenotype A (*rmpA*), which is associated with synthesizing polysaccharide capsules [4]. Additionally, it was demonstrated that the *rmpA* gene was significantly correlated with the hypermucoviscous phenotype and the development of liver abscesses using multivariate analysis [4]. While the etiology is not certain, the reports above clearly illustrate that the gut–liver axis plays a significant role in the development of liver diseases as virulent bacteria in the gut microbiome are correlated with liver damage.

The liver is a vital organ that helps with bile production and metabolization and bilirubin absorption and, most importantly, serves to provide first-pass metabolism to detoxify blood from the digestive system. As excess alcohol, which is consumed by humans or produced by *K. pneumoniae* and other gut bacteria, enters the liver through the bloodstream, it can lead to the formation of reactive oxygen species (ROS), or free radicals. These free radicals provoke the formation of toxic aldehydes, leading to severe liver abscesses. Current treatment methods for liver damage typically involve lifestyle changes, including weight management, abstinence from alcohol, and nutritional therapy. There are also drug therapies, including the administration of corticosteroids to decrease the cytokine response and pentoxifylline to increase 3’,5’-cyclic monophosphate (cAMP) concentrations [5]. However, administration of these drugs may yield more serious side effects, including liver injury and infection [6]. Furthermore, lifestyle changes and current drug therapies do not directly affect the progression of liver damage itself but, rather, they help manage and prevent possible adverse effects, including diabetes and cardiovascular issues. While antibiotic therapy represents a possible direct treatment option to eradicate high alcohol-producing *K. pneumoniae*, many antibiotic resistant *K. pneumoniae* strains have been identified [7]. Moreover, antibiotics lack the specificity to selectively kill harmful *K. pneumoniae* strains and would also eradicate commensal or beneficial bacteria in the human microbiome. Another possible method is the introduction of genetically modified *K. pneumoniae* with reduced virulence [8]; however, this treatment choice risks transferring *K. pneumoniae* genes to other microbes and/or host cells.

Even without genetic modifications, gut-residing microbes can naturally benefit the host. For instance, beneficial microbes may produce antimicrobial agents to combat pathogens, and probiotic bacteria may produce beneficial metabolites to help fight diseases and to alleviate dysbiosis in the microbiome. Probiotics have been defined by the Food and Agriculture Organization of the World Health Organization as live microorganisms that, when administered in sufficient amounts, provide a health benefit for the host [9]. It has been documented that short-chain fatty acids (SCFAs) are produced by these probiotic bacteria. Many SCFAs, such as acetic acid, butyric acid, or propionic acid, have anti-microbial and anti-inflammatory properties. SCFAs are also capable of decreasing inflammation via the inhibition of histone deacetylase (HDAC) [10]. Unlike other treatment options, probiotics do not require host immune activation and may not affect other commensal bacteria. While promising, the Food and Drug Administration (FDA) has not yet approved administering probiotic bacteria for the treatment of liver diseases. Probiotic bacteria in the human gut microbiome can metabolize glucose, fiber, or other prebiotics to produce a beneficial metabolome, a collection of fermentation metabolites including SCFAs [11]. Clinical studies of supplementing gut probiotic bacteria with prebiotics to induce SCFA production for the treatment of human diseases such as type 2 diabetes mellitus have been extensively conducted [12]. The FDA has established definitions for prebiotics and permits food manufacturers to self-affirm GRAS (generally recognized as safe) status for products ultimately labeled as prebiotics [9]. Prebiotics were originally defined by Gibson and Roberfroid in 1995 [13] as “nondigestible food ingredients that beneficially affect the host by selectively stimulating the growth and/or activity of one or a limited number of bacterial species already resident in the colon, and thus attempt to improve host health”. This definition was later revised by Gibson et al [14,15] in 2004 as “a selectively fermented ingredient that allows specific changes, both in the composition and/or activity in the gastrointestinal microflora, that confer benefits upon host well-being and health”. Glucose-based dietary fibers and non-carbohydrate substances including polyunsaturated fatty acid (PUFA) have been used as prebiotics for gut bacteria [16]. The use of prebiotics to naturally manipulate the metabolism of commensal or virulent *K. pneumoniae* may be a non-genetic approach to shift the dysbiotic metabolome to a healthier metabolome in the gut. In this study, we treated commensal *K. pneumoniae* and the *K. pneumoniae* K1 serotype with glucose or linolenic acid. The amounts of alcohol and major SCFAs in the media of *K. pneumoniae* incubated with glucose and linolenic acid were compared.

## 2. Materials and Methods

### 2.1. Bacterial Culture and Fermentation

Chemicals were purchased from Thermo Fisher Scientific (Fair Lawn, NJ, USA) or companies as indicated. A *K. pneumoniae* K1 (1084) strain [4] isolated from patients with liver abscesses was obtained from Dr. Ying–Tsong Chen, Institute of Genomics and Bioinformatics, National Chung Hsing University, Taiwan. A commensal *K. pneumoniae* C strain was isolated from the fecal samples of healthy subjects and identified by 16S ribosomal RNA(rRNA) gene sequencing (Appendix A). *K. pneumoniae* bacteria were cultured in tryptic soy broth (TSB) media to an absorbance at 600 nm [optical density (OD)_600_] of 1.0. The bacterial samples were harvested by centrifugation at 5000× *g* for 10 min. Then, they were washed with and suspended in phosphate buffered saline (PBS). For fermentation, the bacteria (10^5^ colony-forming unit (CFU)/mL) were incubated in rich media (10 g/L yeast extract (Biokar Diagnostics, Beauvais, France), 3 g/L TSB, 2.5 g/L K_2_HPO_4_, and 1.5 g/L KH_2_PO_4_), in the absence or presence of 20 g/L glucose or linolenic acid under anaerobic conditions using the Gas-Pak at 37 °C. Rich media with 20 g/L glucose (G7021) or linolenic acid (L2376) (Sigma, St. Louis, MO, USA) without bacteria were used as controls. The samples were cultured in a Biosafety Level 2 (BSL-2) facility in America Diagnosis Inc., San Diego.

### 2.2. Alcohol Detected by Gas Chromatographic Mass Spectrometry (GC-MS) 

The analysis of the alcohol in each 100 µL sample was conducted in an Agilent 5973 mass selective detector, interfaced to an Agilent 6890 gas chromatography with an Agilent 7694 headspace sampler and a HP 5MS column (25m × 0.25mm × 0.25µm), which were all found at HT Labs Inc., San Diego. The flow rate of the carrier gas helium was set at 1 mL/min. A peak in the GC chromatograph with a retention time of 2.39 min was identified as alcohol and this was further confirmed by MS. Peak areas were used for quantitation. A five-point calibration curve covering the alcohol concentration range of 0–2 g/L was constructed. Alcohol-free samples contained a spike with alcohol at a concentration of 0.05, 0.1, 0.5, 1, and 2 g/L and were analyzed in triplicate.

### 2.3. SCFA Identification by GC-MS 

*K. pneumoniae* K1 or commensal *K. pneumoniae* C (10^5^ CFU/mL) bacteria were incubated in rich media with 2% glucose or linolenic acid for 3 days. The fermentation media was centrifuged at 5000×g to remove bacteria and filtered through 0.2 µm filters. The fermentation media (100 µl) were separately subjected to liquid–liquid extraction with ethyl acetate (1:1, *v*/*v*) for 10 min (Residue Analysis OmniSolv, Millipore, Billerica, MA) after adding an internal standard (0.1 mg/mL ^2^H7-butyric acid, C/D/N Isotopes, Quebec, Canada), followed by acidification with 0.5% ortho-phosphoric acid and saturation with sodium chloride [17,18]. GC-MS analysis for SCFAs in the aqueous phase was performed using an Agilent 5890 Series II GC, coupled with a 5971 MS detector (Agilent Technologies, Inc., Palo Alto, CA) at the Sanford Burnham Prebys Cancer Metabolism Core, San Diego. A 70-eV electron beam was used for ionization. SCFAs in the fermentation media were quantified by a calibration curve made from six non-zero levels using the 500-, 1,000-, 2,000-, 5,000-, and 10,000-fold dilutions of the Free Fatty Acids Test Standard (Restek Corporation, Bellefonte, PA) [19] which contained six SCFAs (acetate, butyrate, propionate, valerate, isobutyrate, and isovalerate). Those SCFAs with higher levels than the background signals were calculated.

### 2.4. Statistical Analysis

Data analysis was performed using a Student *t-*test. The *p*-values of <0.05 (*), <0.01 (**), and <0.001 (***) were considered significant. The mean ± standard deviation (SD) was calculated for at least three independent experiments.

## 3. Results

### 3.1. Alcohol Production in K. pneumoniae

Dietary fibers can be fermentatively converted to glucose by gut bacteria. Both dietary fibers [20] and glucose [21] have been used as substances to promote the SCFA products by the fermentation of gut bacteria. Thus, the ability of glucose (C₆H₁₂O₆) and linolenic acid (C_18_H_30_O_2_) to induce the fermentation of commensal *K. pneumoniae* and the *K. pneumoniae* K1 serotype were examined. The fermentation of *K. pneumoniae* produced detectable alcohol, and its MS spectrum was shown in Figure 1 and concentration was quantified using GC-MS. As shown in Figure 1, in the presence of 20 g/L glucose, the *K. pneumoniae* K1 serotype (149 ± 23 µmol/mL) produced a higher amount of alcohol than the commensal *K. pneumoniae* strain (94 ± 12 µmol/mL). When the carbon source was changed from glucose to linolenic acid, the alcohol production of the *K. pneumoniae* K1 serotype was considerably reduced. This result indicates that linolenic acid can be used as a prebiotic means of reducing the alcohol production of the *K. pneumoniae* K1 serotype.

### 3.2. SCFA Production Enhanced by Linolenic Acid 

Six SCFAs were chosen to establish the calibration curves for the quantification of the SCFAs in the fermentation media of *K. pneumoniae* by GC-MS analysis. As shown in Figure 2, the incubation of commensal *K. pneumoniae* with 20 g/L glucose generated four detectable SCFAs (acetate, propionate, isobutyrate, and butyrate) in the fermentation media. Valerate and isovalerate were not calculated as their levels were lower than the background signals in the GC-MS spectra. The acetate, propionate, isobutyrate, and butyrate were detected in a total ion chromatogram with retention times of 6.0, 8.2, 9.1, and 10.2 min, respectively (Figure 2A). The mass spectra of four detectable SCFAs with major fragments are shown in Figure 2B. The results indicate that commensal *K. pneumoniae* is a fermenting gut bacterium. Although SCFAs were detected following the glucose fermentation of the *K. pneumoniae* K1 serotype, their amounts were markedly lower than those in the glucose fermentation of commensal *K. pneumoniae* (Table 1). Most importantly, the amounts of four major SCFAs (acetate, propionate, isobutyrate, and butyrate) produced by the *K. pneumoniae* K1 serotype were significantly increased when 20 g/L glucose was replaced by 20 g/l linolenic acid (Table 1). For example, 1.0 ± 0.1 or 66 ± 5.0 µM propionate was produced in the media of glucose or linolenic acid fermentation by the *K. pneumoniae* K1 serotype, respectively. Taken together, the data in Figure 1 and Figure 2 demonstrate that linolenic acid can lower alcohol production and increase SCFA production during the fermentation of the *K. pneumoniae* K1 serotype. These data also highlight the ability of linolenic acid to modulate the metabolome of *K. pneumoniae* fermentation.

## 4. Discussion

A well-balanced diet rich in fibers and unsaturated fats leads to an increase in anti-inflammatory bacterial taxa such as *Bifidobacteria* [22]. The data in this study demonstrate for the first time that linolenic acid, a polyunsaturated ω-3 fatty acid, can lower alcohol production yet induce an increase in SCFAs in the fermentation process of the virulent *K. pneumoniae* K1 serotype. Linolenic acid and its derivatives, or linolenic acid-rich foods such as flaxseed, pumpkin, and walnut oils, may function as natural prebiotics to lower the virulence of *K. pneumoniae*. The findings here also suggest that the treatment of liver abscesses with SCFAs may be able to minimize the cytotoxicity induced by high levels of alcohol in the gut. However, SCFAs with relatively short metabolic half-lives [23] may limit their therapeutic effects for the treatment of alcohol-induced liver diseases. Acetate in a mM range was detected in vitro in the fermentation media of *K. pneumoniae*. However, it was known that probiotic intakes may be able to only generate SCFAs in the blood at concentrations in the µM range [24]. Synthesis of an SCFA derivative with a longer half-life may increase the stability of SCFAs in vivo. For example, several propionic acid derivatives have been developed as pharmaceutic aids. These derivatives include drugs with brand names, such as Dermaclens, Transitonyl, Wound Klense, and Andriodermol [25]. SCFAs have been shown to exert crucial physiological effects on several human organs and have anti-inflammatory properties in numerous contexts. For example, butyrate produced by gut microbiota via the gut–brain axis is effective in reducing inflammation and pain in irritable bowel syndrome patients [26]. SCFAs, especially acetate, can significantly reverse ROS production in mesangial cells, and this can be recapitulated by treatment with an agonist for the cognate G protein-coupled receptor (GPCR), free fatty acid receptor 2 (FFAR2) [27]. Furthermore, dietary SCFA (acetate, propionate, and butyrate) intake improves liver metabolism by acting on FFAR3 [28]. Although the literature has indicated that increased levels of propionate promote liver gluconeogenesis in mice [29], results from a clinical study illustrated that propionate supplementation remarkably reduced intrahepatocellular lipid content [30].

*K*. *pneumoniae* is a commensal bacterium in the human gut microbiome, and gut-resident K. *pneumoniae* strains have been shown to possess genotypes similar to those strains detected in liver abscesses and bacteremia [31]. Future works will include sequencing of the genome of *K. pneumoniae* C and comparing the expression of possible virulence genes with the *K. pneumoniae* K1 serotype. Multiple virulence genes in the *K. pneumoniae* K1 serotype significantly contribute to the development of liver abscesses [4]. Although it has been reported that certain gut bacteria can increase blood alcohol levels and induce liver disease [2,3], it is still unclear if commensal or virulent *K. pneumoniae* can yield alcohol via fermentation. Linolenic acid is a PUFA. Results from previous studies have demonstrated that linolenic acid can influence the growth and adhesion of probiotic lactic acid bacterial strains [32]. In addition, molecules with multiple carbon atoms, such as polyethylene glycerol (PEG), have been used as prebiotics to provide carbon sources for bacterial fermentation [18]. Not only intestinal commensal bacteria but also mammalian cells metabolize linolenic acid; these mammalian cells have a unique degradation pathway that yields unique metabolites. After absorption, linolenic acid can be metabolized into eicosapentaenoic acid (EPA), n-3 docosapentaenoic acid (n-3 DPA), and docosahexaenoic acid (DHA) through elongation of the carbon chain in the liver [33]. Thus, it is worth investigating how much SCFAs, EPA, n-3 DPA, and DHA can be produced in the gut when linolenic acid is used as an oral supplement. Although the original definition of a prebiotic was revised in 2004 [14], a panel of experts in microbiology, nutrition, and clinical research was convened by the International Scientific Association for Probiotics and Prebiotics in 2016 to review the definition and scope of prebiotics. The panel updated the definition of a prebiotic as a substrate that is selectively utilized by host microorganisms conferring a health benefit. This definition expands prebiotics to include non-carbohydrate substances including PUFA and conjugated linoleic acid (CLA), applications to body sites other than the gastrointestinal tract, and diverse categories other than food [16]. DHA derived from linolenic acid can exert a positive action by reverting the microbiome composition and increasing the production of anti-inflammatory compounds, like SCFAs, in inflammatory bowel disease [34]. Although our results demonstrated that linolenic acid can be fermented by gut bacteria to trigger the production of beneficial SCFAs in vitro, additional scientific and clinical studies may be required to validate the prebiotic activity of linolenic acid based on the three criteria described by Gibson et al, 2004 [14,15].

Blood alcohol concentration above 11 µmol/mL has been used to predict serum alanine aminotransferase (ALT) and aspartate aminotransferase (AST), two enzymes commonly used as biomarkers for liver damage [35]. Two high alcohol-producing W14 and TH1 strains of *K. pneumoniae* can produce more than 30 µmol/mL alcohol in a 12 h in vitro culture [2]. The results of this study showed that glucose fermentation of the *K. pneumoniae* K1 serotype for 3 days yielded approximately 149 µmol/mL alcohol in the media, suggesting that the *K. pneumoniae* K1 serotype is a high alcohol-producing strain. It has been documented that alcohol consumption can cause a lower abundance of *Bacteroidetes* and a higher abundance of *Proteobacteria* in the human gut microbiome [36]. Thus, the interaction of alcohol-producing *K. pneumoniae* with *Bacteroidetes* and *Proteobacteria* can be explored. Enzymes in the pathway of the glucose fermentation of gut bacteria have been well characterized [37]. However, the enzymes involved in the conversion of linolenic acid to alcohol during the fermentation of *K. pneumoniae* have not been identified. Furthermore, the in vivo efficacy of using linolenic acid as a supplement to ameliorate alcohol-induced liver damage has not yet been determined.

The primary complication of using antibiotic therapy to eliminate virulent *K. pneumoniae* is the possibility of developing resistant bacterial strains. The data in this study emphasize the significance of using linolenic acid as a potential prebiotic for the production of beneficial metabolites during bacterial fermentation. Vaccines for new microbial strains cannot be administered on time during outbreaks. Humans host tremendous amounts of commensal bacteria in the microbiome at all times. Boosting the fermentation activity of these commensal bacteria by using “prebiotics” to augment a beneficial metabolome directly provides a readily available solution for the treatment of infections. It is estimated that there are five million annual cases of alcoholic liver diseases in the United States [38], with a mortality rate ranging from 30% to 50% [39]. Using prebiotics as supplements may impact these patients by shaping the metabolome of the virulent *K. pneumoniae* by promoting the production of beneficial SCFAs instead of alcohol.

## Figures and Tables

**Figure 1 microorganisms-08-00773-f001:**
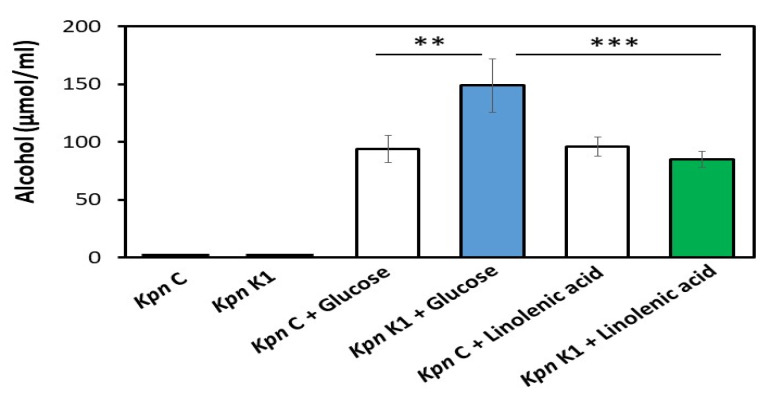
Alcohol production from glucose or linolenic acid fermentation of commensal *K. pneumoniae* (*Kpn*) C and virulent *Kpn* K1 strain. *Kpn* C or *Kpn* K1 bacteria (10^5^ CFU/mL) were incubated in rich media in the absence or presence of 20 g/L glucose or linolenic acid for 3 days. The alcohol in the fermentation media was detected by HP6890 GC-MS with a retention time of 2.39 min. Alcohol concentrations (µmol/mL) in rich media of *Kpn* C alone, *Kpn* K1 alone, *Kpn C* with glucose, *Kpn K1* with glucose, *Kpn C* with linolenic acid, and *Kpn K1* with linolenic acid were determined. The mean ± SD from three independent experiments and *p*-values of < 0.01 (**) and < 0.001 (***) via a Student *t*-test were calculated and denoted accordingly.

**Figure 2 microorganisms-08-00773-f002:**
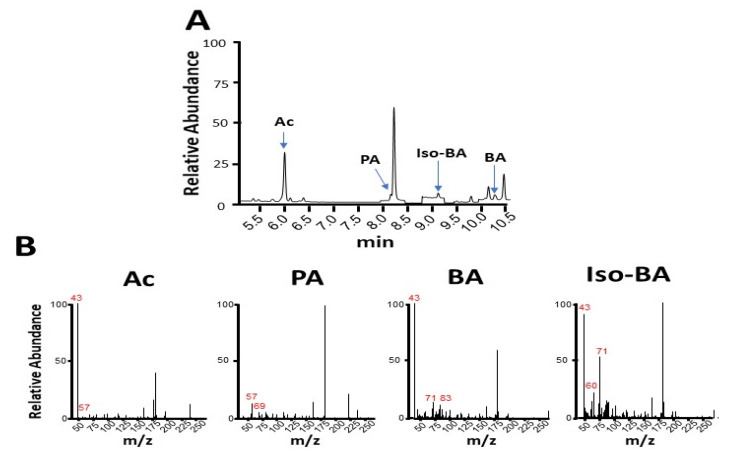
Production of short-chain fatty acids (SCFAs) by glucose fermentation of *K. pneumoniae* C strain. (**A**) Total ion chromatogram for separation of the mixture of SCFAs containing acetate (Ac), propionate (PA), isobutyrate (iso-BA), and butyrate (BA) was displayed by running the GC-MS analysis. (**B**) The mass spectra for acetate, propionate, isobutyrate, and butyrate with their corresponding molecular ions (m/z) are shown.

**Table 1 microorganisms-08-00773-t001:** Enhancement of SCFA production of *K. pneumoniae* (*Kpn*) K1 by linolenic acid fermentation.

SCFAs	*Kpn* C /Glucose	*Kpn* C /Linolenic acid	*Kpn* K1 /Glucose	*Kpn* K1 / Linolenic acid
Acetate (mM)	1.3 ± 0.1	$2.8^{\;*}\;\pm \;0.3$	0.6 ± 0.02	$9.0{\;}^{**}\;\pm \;1.1$
Propionate (µM)	6.0 ± 0.7	$15.0{\;}^{*}\;\pm \;2.0$	1.0 ± 0.1	$66.0^{\;***}\;\pm \;5.0$
Isobutyrate (µM)	20.9 ± 3.2	22.5 ± 4.5	18.5 ± 3.3	$24.5{\;}^{*}\;\pm \;4.2$
Butyrate (µM)	25.1 ± 4.5	27.2 ± 4.2	21.5 ± 3.8	$30.1{\;}^{**}\;\pm \;4.2$

The levels of acetate, propionate, isobutyrate, and butyrate produced by glucose and linolenic acid fermentation of *Kpn* C and *Kpn* K1 strains are shown. Data depict the mean ± SD of three separate experiments. The *p*-values of < 0.05 (*), < 0.01 (**), and < 0.001 (***) via a Student *t*-test (*Kpn* C/glucose vs. *Kpn* C/linolenic acid; *Kpn* K1/glucose vs. *Kpn* K1/linolenic acid) are denoted.

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
