# Peer review of "Manipulation of Alcohol and Short-Chain Fatty Acids in the Metabolome of Commensal and Virulent Klebsiella pneumoniae by Linolenic Acid"

_microorganisms, 2020, doi:10.3390/microorganisms8050773_

Round 1
Reviewer 1 Report
Line 14. Please avoid the use of “we, our etc.”. Please correct it in the whole manuscript.
Lines 60-69. Please provide references for this part. It is very important to support this information.
Lines 97-98. Please avoid the use of % and instead use g/l, in order to have uniform results.
Lines 101-109. Please provide more details in gc/ms analysis. Any treatment of the sample? Why GC/MS was used since it is more usual to use a simple GC. The sample was directly injected or only the headspace? If the headspace was selected, I am wondering if it is appropriate for quantitative analysis. As I mentioned before the simple GC analysis of a directly injected sample is more usual and appropriate. If necessary provide references.
Lines 110-121. Please provide more details for extraction and GC/MS analysis. If necessary provide references.
Lines 128-132. Please correct. Extensive use of italics.
Lines 140-141. Please use italics for microorganisms.
Lines 142-143. This is a well-known information about the identification in a MS detector. Delete.
Figure 1. Delete A, B, C. All of them represent a well-known GC/MS analysis. In addition, in D it is better to use μmol.
Line 161. Please explain BAC.
Lines 181-185. This is a well-known information about the identification in a MS detector. Delete.
Line 190. Correct propoinate to propionate
Figure 2. Delete. Well-known information.
Table 2. Please revise table title. Some information for example p values may be presented below table.
Lines 221-222. Irrelevant. Delete.
Lines 228-229. Please provide information about the third author SM
Author Response
April 30, 2020
Point-by-Point Response to Reviewers
Thanks for reviewing our manuscript (microorganisms-784697) entitled “Prebiotic Manipulation of Alcohol and Short-chain Fatty Acids in the Metabolome of Commensal and Virulent Klebsiella pneumoniae”. We have made revision based on reviewers’ comments. Changes have been underlined in the revised manuscript. Below are the point-by-point our address to reviewers’ comments.
Editor’ comments:
Question 1. Although the manuscript has novelty and may be interested to readers in terms of gut-liver axis, the following improvements in the revised manuscript are needed.
(1). The method of quantification of SCFAs in mass spectrometry should be described in detail. Why only 4 SCFAs can be detected? How about other SCFAs and metabolites? The concentrations of detected SCFAs at mM are high. Authors need to compare other previous studies to know the concentrations of SCFAs detected in both gut and blood.
Answers:
The answers have been added into the Sections of Methods, Results, and Discussion as follows. One new reference [32] has been cited.
“SCFAs in the fermentation media were quantified by a calibration curve made from six non-zero levels using the 500-, 1,000-, 2,000-, 5,000-, and 10,000-fold dilution of the Free Fatty Acids Test Standard (Restek Corporation, Bellefonte, PA) [16] which contains six SCFAs (acetate, butyrate, propionate, valerate, isobutyrate, and isovalerate). Those SCFAs with higher levels than the background signals were calculated.”
“Six SCFAs were chosen to establish the calibration curves for quantification of SCFAs in fermentation media of K. pneumoniae by GC-MS analysis. As shown in Figure 2, incubation of commensal K. pneumoniae with 20 g/l glucose generated four detectable SCFAs (acetate, propionate, isobutyrate, and butyrate) in fermentation media. Valerate and isovalerate were not calculated as their levels were lower than the background signals in GC-MS spectra. The acetate, propionate, isobutyrate, and butyrate were detected in a total ion chromatogram with at retation time of 6.0, 8.2, 9.1, 10.2 min, respectively (Figure 2A).”
“Acetate in a mM range was detected in vitro in the fermentation media of K. pneumoniae (Table 1). However, it was known that probiotic intakes may be able to only generate SCFAs in the blood at concentrations in the µM range [32].”
32. Parker, A.; Fonseca, S.; Carding, S.R. Gut microbes and metabolites as modulators of blood-brain barrier integrity and brain health. Gut Microbes 2020, 11, 135-157. doi: 10.1080/19490976.2019.1638722.
Question 2. (2). The term of "prebiotic" (eg glucose) should be well defined. How those prebiotics will be metabolized or degraded if giving them as oral supplements.
Answers:
The prebiotic definition by FDA has been added under the Section of Introduction. The metabolism of linolenic acid has been described in the Section of Discussion. Three new references [9, 13, 21] have been cited.
“Probiotics have been defined by the Food and Agriculture Organization of the World Health Organization as live microorganisms that, when administered in sufficient amounts, provide a health benefit on the host [9].”
“FAD has established definitions for prebiotics and permits food manufacturers to self-affirm GRAS (Generally Recognized as Safe) status for products ultimately labeled as prebiotics [9]. The National Institutes of Health (NIH) envisions that prebiotics are a part of a ‘biologically-based’ group of foods that also include botanicals, animal-derived extracts, vitamins, minerals, fatty acids, amino acids, proteins, whole diets, and functional foods [13].”
“Not only intestinal commensal bacteria but also mammalian cells metabolize linolenic acid; these mammalian cells have a unique degradation pathway that yields unique metabolites. After absorption, linolenic acid can be metabolized into eicosapentaenoic acid (EPA), n-3 docosapentaenoic acid (n-3 DPA) and docosahexaenoic acid (DHA) through elongation of the carbon chain [21]. Thus, it is worth investigating how much SCFAs, EPA, n-3 DPA and DHA can be produced in the gut when linolenic acid is used as an oral prebiotic.”
9. Venugopalan, V.; Shriner, K.A.; Wong-Beringer, A. Regulatory oversight and safety of probiotic use. Emerg Infect Dis 2010, 16, 1661-5. doi: 10.3201/eid1611.100574.
13. Hutkins, R.W.; Krumbeck, J.A.; Bindels, L.B.; Cani, P.D.; Fahey, G. Jr.; Goh, Y.J.; Hamaker, B.; Martens, E.C.; Mills, D.A.; Rastal, R.A. et al. Prebiotics: why definitions matter. Curr Opin Biotechnol 2016, 37, 1-7. doi: 10.1016/j.copbio.2015.09.001.
21. Nagatake T, Kunisawa J. Emerging roles of metabolites of ω3 and ω6 essential fatty acids in the control of intestinal inflammation. Int Immunol. 2019 Aug 23;31(9):569-577. doi: 10.1093/intimm/dxy086. PMID: 30722032
Question 3. (3) The term of "commensal" should be also clearly defined. How readers can know the commensal authors identified is really commensal. Authors should include other commensal strains that have been published.
Answers:
This has been discussed in the Section of Discussion. Also see reference [18].
“K. pneumoniae is a commensal bacterium in the human gut microbiome, and gut-resident K. pneumoniae strains have been shown to possess genotypes similar to those strains detected in liver abscess and bacteremia [18]. Future works will include sequencing of the genome of the K. pneumoniae C and comparing the expression of possible virulence genes with K. pneumoniae K1 serotype.”
Reviewer 1’ comments:
Question 4. Line 14. Please avoid the use of “we, our etc.”. Please correct it in the whole manuscript.
Lines 60-69. Please provide references for this part. It is very important to support this information.
Lines 97-98. Please avoid the use of % and instead use g/l, in order to have uniform results.
Answers:
The word of “we, our etc” has been replaced with “in this study or others”.
References [6, 7, 8] for Line 60-69 have been cited.
2% glucose or linolenic acid has been changed to 20 g/l.
6. Vargas, J.I; Arrese, M.; Shah, V.H.; Arab, J.P. Use of Statins in Patients with Chronic Liver Disease and Cirrhosis: Current Views and Prospects. Curr Gastroenterol Rep 2017, 19, 43. doi: 10.1007/s11894-017-0584-7.
7. Navon-Venezia, S.; Kondratyeva, K.; Carattoli, A. Klebsiella pneumoniae: a major worldwide source and shuttle for antibiotic resistance. FEMS Microbiol Rev 2017, 41, 252-275. doi: 10.1093/femsre/fux013.
8. Jayol, A.; Poirel, L.; Villegas, M.V.; Nordmann, P. Modulation of mgrB gene expression as a source of colistin resistance in Klebsiella oxytoca. Int J Antimicrob Agents 2015, 46, 08-10. doi: 10.1016/j.ijantimicag.2015.02.015.
Question 5. Lines 101-109. Please provide more details in gc/ms analysis. Any treatment of the sample? Why GC/MS was used since it is more usual to use a simple GC. The sample was directly injected or only the headspace? If the headspace was selected, I am wondering if it is appropriate for quantitative analysis. As I mentioned before the simple GC analysis of a directly injected sample is more usual and appropriate. If necessary provide references.
Lines 110-121. Please provide more details for extraction and GC/MS analysis. If necessary provide references.
Answers:
Also see Answers for Editor’s Question 1. Two new references [14, 15] were cited.
Detailed information of GC-MS analysis has been added. No headspace was used in GC-MS for SCFA detection.
Reference an additional information of the protocol of sample extraction have been added.
Dr. David A Scott for conduction of GC-MS at the Sanford Burnham Prebys Cancer Metabolism Core, San Diego has been acknowledged.
“The fermentation media was centrifuged at 5,000 x g to remove bacteria and filtered through 0.2 µm filters. The fermentation media (100 µl) were separately subjected to liquid-liquid extraction with ethyl acetate (1:1, v/v) for 10 min (Residue Analysis OmniSolv, Millipore, Billerica, MA) after adding an internal standard (0.1 mg/ml 2H7-butyric acid, C/D/N Isotopes, Quebec, Canada), followed by acidification with 0.5% ortho-phosphoric acid and saturation with sodium chloride [14, 15]. GC-MS analysis for SCFAs in the aqueous phase was performed using an Agilent 5890 Series II GC, coupled with a 5971 MS detector (Agilent Technologies, Inc., Palo Alto, CA) at the Sanford Burnham Prebys Cancer Metabolism Core, San Diego.”
“We thank Dr. David A Scott at the Sanford Burnham Prebys Cancer Metabolism Core, San Diego for GC-MS analysis.”
Question 6. Lines 128-132. Please correct. Extensive use of italics.
Lines 140-141. Please use italics for microorganisms.
Lines 142-143. This is a well-known information about the identification in a MS detector. Delete.
Answers:
Errors for italics in couple sentences and for microorganisms in the whole manuscript have been corrected.
The well-known information of chemical structure for each iron fragment in MS spectra has been deleted.
Question 7. Figure 1. Delete A, B, C. All of them represent a well-known GC/MS analysis. In addition, in D it is better to use μmol.
Line 161. Please explain BAC.
Lines 181-185. This is a well-known information about the identification in a MS detector. Delete.
Answers:
A, B, C in Figure 1 have been deleted. The label of µmole/ml has been changed to µmol/ml.
BAC for “Blood alcohol concentration” has been spelled out.
The well-known information of chemical structure for each iron fragment in MS spectra has been deleted
Question 8. Line 190. Correct propoinate to propionate.
Figure 2. Delete. Well-known information.
Table 2. Please revise table title. Some information for example p values may be presented below table.
Lines 221-222. Irrelevant. Delete.
Lines 228-229. Please provide information about the third author SM.
Answers:
The wrong spelling of propionate has been corrected.
Well-known chemical structure in Figure 2 has been deleted.
Title for Table 1 has been revised. Information has been moved below table.
Irrelevant sentence with original reference 21 has been deleted.
Contribution of the third author has been added.
“S.M. provided knowledge of gut microbiome and edited the manuscript.”
Reviewer 2’ comments:
Question 9. The manuscript entitled: “Prebiotic Manipulation of Alcohol and Short-chain Fatty Acids in the Metabolome of Commensal and Virulent Klebsiella pneumoniae” is a reviewed about the effects of prebiotics over alcohol. The topic is interesting due to the relationship between microbiota and gut liver axis, however more revision is required according to follow.
Introduction: The terms probiotics and prebiotics should be defined. Also, some example of their used in gut microbiota axis should be indicate. In fact, previous researches about the effect of both in SCFAs have to be cited.
Answers:
Also see answers to Editor’s question 2 with references [9, 13, 21]. Prebiotic has been defined. Definition of “Probiotic” with reference [9] and previous researches about the effects of SCFAs from probiotic/prebiotic (reference 12) have been added under Section of Introduction.
“Probiotics have been defined by the Food and Agriculture Organization of the World Health Organization as live microorganisms that, when administered in sufficient amounts, provide a health benefit on the host [9].”
“Clinical studies of supplementing gut probiotic bacteria with prebiotics to induce SCFA production for treatment of human diseases such as type 2 diabetes mellitus have been extensively conducted [12].”
12. Tonucci, L.B.; Olbrich Dos Santos, K.M.; Licursi de Oliveira, L.; Rocha Ribeiro, S.M.; Duarte Martino, H.S. Clinical application of probiotics in type 2 diabetes mellitus: A randomized, double-blind, placebo-controlled study. Clin Nutr 2017, 36, 85-92. doi: 10.1016/j.clnu.2015.11.011.
Question 10. Method: Number of samples and prebiotics analysed are missing.
Answers:
Number of samples and prebiotics analysed has been added in the Section of Method.
“20 g/l glucose (G7021) or linolenic acid (L2376) (Sigma)”
100 µl samples for GC-MS analysis.
Question 11. Results and Discussion: I think that both should be in separate sections.
Discussion is not being enough. Implications of the effects of prebiotics through SCFAs in other organs like the brain should be indicated.
Also the effects of alcohol over gut microbiota bacterial should be indicated.
Could be interesting to cite the role or the significance of prebiotics analysed over NAFLD.
Answers:
Results and Discussion has been separated.
The effects of prebiotics through SCFAs in brain has been described in the Section of Discussion. (Reference 25)
The alcohol effects on gut microbiota have been stated in the Section of Discussion. (Reference 23)
The significance of prebiotics analysed over NAFLD has been described in the Section of Discussion. (References 34, 35)
“SCFAs have been shown to exert crucial physiological effect on serval organs and have anti-inflammatory properties in numerous contexts. For example, butyrate produced by gut microbiota via the gut-brain axis is effective in reducing inflammation and pain in irritable bowel syndrome [25].”
“It has been documented that alcohol consumption can cause a lower abundance of Bacteroidetes and a higher abundance of Proteobacteria in the human gut microbiome [23]. Thus, the interaction of alcohol-producing K. pneumoniae with Bacteroidetes and Proteobacteria can be explored.”
“It is estimated that there are 5 million annual cases of alcoholic liver diseases in the United States [34], with a mortality rate ranging from 30 to 50% [35]. Using prebiotics as supplements may impact these patients by shaping the metabolome of the virulent K. pneumoniae by promoting the production of beneficial SCFAs instead of alcohol.”

Reviewer 2 Report
The manuscript entitled: “Prebiotic Manipulation of Alcohol and Short-chain Fatty Acids in the Metabolome of Commensal and Virulent Klebsiella pneumoniae” is a reviewed about the effects of prebiotics over alcohol. The topic is interesting due to the relationship between microbiota and gut liver axis, however more revision is required according to follow.
Introduction
- The terms probiotics and prebiotics should be defined. Also, some example of their used in gut microbiota axis should be indicate. In fact, previous researches about the effect of both in SCFAs have to be cited.
Method:
- Number of samples and prebiotics analysed are missing
Results and Discussion:
- I think that both should be in separate sections.
- Discussion is not being enough. Implications of the effects of prebiotics through SCFAs in other organs like the brain should be indicated.
- Also the effects of alcohol over gut microbiota bacterial should be indicated.
- Could be interesting to cite the role or the significance of prebiotics analysed over NAFLD
Author Response

(The authors gave the same response as above.)

Round 2
Reviewer 1 Report
The authors did not provide the definition of prebiotics. I do not think that lines 83-90 is any definition of prebiotics. In addition, the authors added in lines 87-90 something that may also cause misunderstandings. “The National Institutes of Health (NIH) envisions that prebiotics are a part of a ‘biologically-based’ group of foods that also include botanicals, animal derived extracts, vitamins, minerals, fatty acids, amino acids, proteins, whole diets, and functional foods [13].” This must be deleted since one may consider that prebiotic are also minerals, vitamins etc that is also totally wrong.
Please see the following
The original definition was introduced in 1995 “non-digestible food ingredient that beneficially affects the host by selectively stimulating the growth and/or activity of one or a limited number of bacteria already resident in the colon” Gibson and Roberfroid 1995.
This definition was updated in 2004, and prebiotics are now defined as ‘‘selectively fermented ingredients that allow specific changes, both in the composition and/or activity in the GI microflora that confer benefits upon host wellbeing and health’’ (Gibson et al. 2004).
The three criteria required for a prebiotic effect are as follows (Gibson et al. 2004):
- Resistant to gastric acidity and hydrolysis by mammalian enzymes and GI absorption
- Can be fermented by intestinal microflora
- Selectively stimulates the growth and/or activity of intestinal bacteria associated with health and wellbeing.
Please use the above mentioned ref to provide the definition.
In addition, according to authors glucose is a prebiotic. However, according to the above mentioned definitions of prebiotics this is not true.
In the context of the digestive tract, this means that a prebiotic substrate cannot be absorbed in the small intestine (excludes, e.g., glucose, sucrose, fructose, lactose, or even the proportion of starch that is hydrolysed to produce glucose), and must persist into the colon, where it can deliver health benefits.
Lines 145-147. “Dietary fibers can be fermentatively converted to glucose by gut bacteria. Both dietary fibers and glucose have been classified as prebiotics to promote the SCFA products by fermentation of gut bacteria [17].” Please provide the exact text from the reference that stated that. I am totally sure that glucose is not a prebiotic as I mentioned above. In addition, in the ref 17 the authors write about Glucose-based prebiotic fibers and not glucose.
Please also provide the appropriate references to prove that linolenic acid is a prebiotic. In addition, explain if the linolenic acid fulfills all the above mentioned 3 criteria to be classified as prebiotic (Gibson et al. 2004).
In my opinion the work is not accurate enough to support so “large” conclusions. The experiments are limited in simple synthetic medium fermentations.
Gibson, G. R., Probert, H. M., van Loo, J. A. E., Rastall, R. A. and Roberfroid, M. B. 2004. Dietary modulation of the human colonic microbiota: updating the concept of prebiotics. Nutrition Research Reviews 17: 259-275.
Gibson, G. R. and Roberfroid, M. B. 1995. Dietary modulation of the human colonic microflora introducing the concept of probiotics. Journal of Nutrition 125: 1401-1412.
Gibson, G. R., Scott, K. P., Rastall, R. A., Tuohy, K. M., Hotchkiss, A., Dubert-Ferrandon, A., ... & Macfarlane, S. (2010). Dietary prebiotics: current status and new definition. Food Sci Technol Bull Funct Foods, 7(1), 1-19.
Marinaki, et al., 2016. Probiotic yogurt production with Lactobacillus casei and prebiotics. Current Research in Nutrition and Food Science Journal, 4(Special Issue Nutrition in Conference October 2016), 48-53.http://dx.doi.org/10.12944/CRNFSJ.4.Special-Issue-October.07
Author Response
May 17, 2020
Point-by-Point Response to Reviewers
Thanks for reviewing our revised manuscript (microorganisms-784697) titled “Manipulation of Alcohol and Short-chain Fatty Acids in the Metabolome of Commensal and Virulent Klebsiella pneumonia by linolenic acid”. We have made revisions based on reviewers’ comments. Changes have been underlined in the revised manuscript. Below are our point-by-point address to reviewers’ comments.
Reviewer 1’ comments:
Question 1. The authors did not provide the definition of prebiotics. I do not think that lines 83-90 is any definition of prebiotics. In addition, the authors added in lines 87-90 something that may also cause misunderstandings. “The National Institutes of Health (NIH) envisions that prebiotics are a part of a ‘biologically-based’ group of foods that also include botanicals, animal derived extracts, vitamins, minerals, fatty acids, amino acids, proteins, whole diets, and functional foods [13].” This must be deleted since one may consider that prebiotic are also minerals, vitamins etc that is also totally wrong.
Please see the following
The original definition was introduced in 1995 “non-digestible food ingredient that beneficially affects the host by selectively stimulating the growth and/or activity of one or a limited number of bacteria already resident in the colon” Gibson and Roberfroid 1995.
This definition was updated in 2004, and prebiotics are now defined as ‘‘selectively fermented ingredients that allow specific changes, both in the composition and/or activity in the GI microflora that confer benefits upon host wellbeing and health’’ (Gibson et al. 2004).
The three criteria required for a prebiotic effect are as follows (Gibson et al. 2004):
- Resistant to gastric acidity and hydrolysis by mammalian enzymes and GI absorption
- Can be fermented by intestinal microflora
- Selectively stimulates the growth and/or activity of intestinal bacteria associated with health and wellbeing.
Please use the above mentioned ref to provide the definition.
Answers:
The definition of prebiotic by NIH in original reference 13 has been deleted. As reviewer’s suggested, two prebiotic definitions by Gibson and Roberfroid 1995 and Gilbson et al., 2004 have been included. Furthermore, we also include an updated definition of prebiotic published in Nat Rev Gastroenterol Hepatol2017, 14, 491-502 with a title of “Expert consensus document: The International Scientific Association for Probiotics and Prebiotics (ISAPP) consensus statement on the definition and scope of prebiotics”.
“The panel updated the definition of a prebiotic: a substrate that is selectively utilized by host microorganisms conferring a health benefit. This definition expands the concept of prebiotics to possibly include non-carbohydrate substances, applications to body sites other than the gastrointestinal tract, and diverse categories other than food.”
Linolenic acid is a polyunsaturated fatty acid (PUFA). Although PUFA has been included as a non-carbohydrate prebiotic by updated definition, we agree reviewer 1’s comments on the prebiotic property of linolenic acid. The term of prebiotic for linolenic acid in whole manuscript has been carefully used. Additional scientific and clinical studies are needed to validate the prebiotic activity of linolenic acid.
The word of “prebiotic” in the original title has been deleted. The new title is “Manipulation of Alcohol and Short-chain Fatty Acids in the Metabolome of Commensal and Virulent Klebsiella pneumoniae by Linolenic Acid”.
Question 2. In addition, according to authors glucose is a prebiotic. However, according to the above mentioned definitions of prebiotics this is not true.
In the context of the digestive tract, this means that a prebiotic substrate cannot be absorbed in the small intestine (excludes, e.g., glucose, sucrose, fructose, lactose, or even the proportion of starch that is hydrolysed to produce glucose), and must persist into the colon, where it can deliver health benefits.
Lines 145-147. “Dietary fibers can be fermentatively converted to glucose by gut bacteria. Both dietary fibers and glucose have been classified as prebiotics to promote the SCFA products by fermentation of gut bacteria [17].” Please provide the exact text from the reference that stated that. I am totally sure that glucose is not a prebiotic as I mentioned above. In addition, in the ref 17 the authors write about Glucose-based prebiotic fibers and not glucose.
Answers:
Glucose is no longer written as a prebiotic in the revised manuscript.
The first sentence in the Result Section (original Line 145-147) has been re-written as below. A new reference [21] has been cited.
“Dietary fibers can be fermentatively converted to glucose by gut bacteria. Both dietary fibers [20] and glucose [21] have been used as substances to promote the SCFA products by fermentation of gut bacteria. Thus, the ability of glucose (C₆H₁₂O₆) and linolenic acid (C18H30O2) to induce fermentation of commensal K. pneumoniae and K. pneumoniae K1 serotype were examined.”
- Egert, M.; de Graaf, A.A.; Maathuis, A.; de Waard, P.; Plugge, C.M.; Smidt, H.; Deutz, N.E.; Dijkema, C.; de Vos, W.M.; Venema, K. Identification of glucose-fermentingbacteriapresent in an in vitro model of the human intestine by RNA-stable isotope probing. FEMS Microbiol Ecol 2007, 60, 126-135. doi: 10.1111/j.1574-6941.2007.00281.x.
Question 3. Please also provide the appropriate references to prove that linolenic acid is a prebiotic. In addition, explain if the linolenic acid fulfills all the above mentioned 3 criteria to be classified as prebiotic (Gibson et al. 2004).
In my opinion the work is not accurate enough to support so “large” conclusions. The experiments are limited in simple synthetic medium fermentations.
Gibson, G. R., Probert, H. M., van Loo, J. A. E., Rastall, R. A. and Roberfroid, M. B. 2004. Dietary modulation of the human colonic microbiota: updating the concept of prebiotics. Nutrition Research Reviews 17: 259-275.
Gibson, G. R. and Roberfroid, M. B. 1995. Dietary modulation of the human colonic microflora introducing the concept of probiotics. Journal of Nutrition 125: 1401-1412.
Gibson, G. R., Scott, K. P., Rastall, R. A., Tuohy, K. M., Hotchkiss, A., Dubert-Ferrandon, A., ... & Macfarlane, S. (2010). Dietary prebiotics: current status and new definition. Food Sci Technol Bull Funct Foods, 7(1), 1-19.
Marinaki, et al., 2016. Probiotic yogurt production with Lactobacillus casei and prebiotics. Current Research in Nutrition and Food Science Journal, 4(Special Issue Nutrition in Conference October 2016), 48-53.http://dx.doi.org/10.12944/CRNFSJ.4.Special-Issue-October.07
Answers:
An updated definition of prebiotic has been published in Nat Rev Gastroenterol Hepatol 2017, 14, 491-502. The polyunsaturated fatty acid (PUFA) has been recognized as a non-carbohydrate prebiotic. Furthermore, our results have demonstrated that linolenic acid can be fermented by gut bacteria and triggered the production of beneficial short-chain fatty acids (SCFAs). However, we totally agree to reviewer’s comments that additional research is needed to validate the prebiotic activity of linolenic acid based on three criteria made by Gibson et al, 2004.
The paragraph below has been added into Discussion Section.
“DHA derived from linolenic acid exerted a positive action by reverting the microbiome composition and increasing the production of anti-inflammatory compounds, like SCFAs, in inflammatory bowel disease [35]. Although our results demonstrated that linolenic acid can be fermented by gut bacteria to trigger the production of beneficial SCFAs in vitro, additional scientific and clinical studies may be required to validate the prebiotic activity of linolenic acid based on the three criteria described by Gibson et al, 2004 [14, 15].”
References from Gibson GR’s groups have been cited in the review manuscript.
Reviewer 2’ comments:
Question 4. The recent version of the manuscript has improved considerably. There are only a few minor questions
Introduction
- The term of prebiotics is still well not defined.
- Line 85: FAD?
Answers:
Also see answer to Question 1 from reviewer 1 in terms of prebiotic definition.
Typo on FAD has been corrected.
An updated definition of prebiotic has published in Nat Rev Gastroenterol Hepatol 2017, 14, 491-502.
The polyunsaturated fatty acid (PUFA) which linolenic acid belongs to has been included as a non-carbohydrate prebiotic based on the updated definition of prebiotic.
“Prebiotics were originally defined by Gibson and Roberfroid in 1995 [13] as ‘nondigestible food ingredients that beneficially affect the host by selectively stimulating the growth and/or activity of one or a limited number of bacterial species already resident in the colon, and thus attempt to improve host health’. This definition was later revised by Gibson et al [14, 15] in 2004 as ‘a selectively fermented ingredient that allows specific changes, both in the composition and/or activity in the gastrointestinal microflora, that confer benefits upon host well-being and health’. Glucose-based dietary fibers and non-carbohydrate substances including polyunsaturated fatty acid (PUFA) have been used as prebiotics for gut bacteria [16].”
Question 5. Discussion:
- I think that both should be in separate sections.
- Division of results-discussion has enhanced the manuscript. However, the discussion still needs to be improved because is a little confused. Discussion should start with the main results of their data (line 282). After that, they have to discuss their data according to the literature. The reference to the table (line 284 and 290) should be eliminated to the discussion section.
Answers:
Discussion has been rearranged based on reviewer’s comment. Discussion started with main results of our data at linolenic acid fermentation and SCFA production. After that, we discussed the metabolites (alcohol, SCFA and others) of linolenic acid fermentation of gut bacteria according to the literature. Updated definition of prebiotic and the linolenic acid as a potential prebiotic have been also fully discussed based on recent literature.
Reviewer 2 Report
The recent version of the manuscript has improved considerably. There are only a few minor questions
Introduction
- The term of prebiotics is still well not defined.
- Line 85: FAD?
Discussion:
- I think that both should be in separate sections.
- Division of results-discussion has enhanced the manuscript. However, the discussion still needs to be improved because is a little confused. Discussion should start with the main results of their data (line 282). After that, they have to discuss their data according to the literature. The reference to the table (line 284 and 290) should be eliminated to the discussion section.
Author Response

(The authors gave the same response as above.)

Round 3
Reviewer 1 Report
The manuscript has been improved.
Especially the addition of ref Gibson, G.R.; Hutkins, R.; Sanders, M.E.; Prescott, S.L.; Reimer, R.A.; Salminen, S.J.; Scott, K.; Stanton, C.; Swanson, K.S.; Cani, P.D., et al. Expert consensus document: The International Scientific Association for Probiotics and Prebiotics (ISAPP) consensus statement on the definition and scope of prebiotics. Nat Rev Gastroenterol Hepatol 2017, 14, 491-502. doi: 10.1038/nrgastro.2017.75. provides to manuscript the appropriate background for the aim of the research.
In my opinion the manuscript may be accepted for publication.